# Training Pansharpening Networks at Full Resolution Using Degenerate Invariance

## ABSTRACT

Pan-sharpening is an important technique for remote sensing imaging systems to obtain high resolution multispectral images. Existing deep learning-based methods mostly rely on using pseudo-groundtruth multi-spectral images for supervised learning. The whole training process only remains at the scale of reduced resolution, which means that the impact of the degradation process is ignored and high-quality images cannot be guaranteed at full resolution. To address the challenge, we propose a new unsupervised framework that does not rely on pseudo-groundtruth but uses the invariance of the degradation process to build a consistent loss function on the original scale for network training. Specifically, first, we introduce the operator learning method to build an exact mapping function from multi-spectral to panchromatic images and decouple spectral features and texture features. Then, through joint training, operators and convolutional networks can learn the spatial degradation process and spectral degradation process at full resolution, respectively. By introducing them to build consistency constraints, we can train the pansharpening network at the original full resolution. Our approach could be applied to existing pansharpening methods, improving their usability on original data, which is matched to practical application requirements. The experimental results on different kinds of satellite datasets demonstrate that the new network outperforms state-of-the-art methods both visually and quantitatively.

## CCS CONCEPTS

• **Computing methodologies → Image processing**.

## KEYWORDS

Unsupervised Training Framework; Pan-sharpening Method; Operator Learning

## 1 INTRODUCTION

Remote sensing technology has rapidly advanced alongside growing satellite data volumes, enabling progress in fields like agriculture and environmental monitoring [47, 49]. Current satellite imaging systems commonly feature multispectral sensors, allowing observation across multiple wavelengths. However, achieving sufficient signal-to-noise within the mechanical constraints of multispectral sensor design necessitates a certain instantaneous field of view,

*ACM MM, 2024, Melbourne, Australia*
© 2024 Copyright held by the owner/author(s). Publication rights licensed to ACM.
ACM ISBN 978-x-xxxx-xxxx-x/YY/MM
https://doi.org/10.1145/nnnnnnn.nnnnnnn

**Unpublished working draft. Not for distribution.**

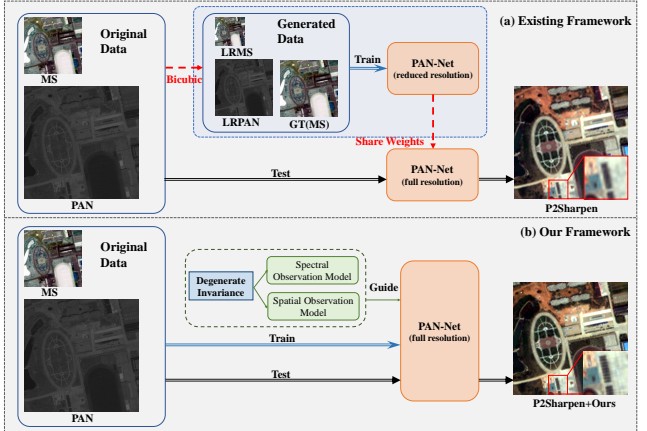

**Figure 1: Comparison of our pansharpening training framework with existing one. (a) Existing PAN-Net supervised training methods choose to generate data containing ground-truth at reduced resolution, then they apply the network trained at reduced resolution directly to the original scale. However, there are some artificial priors in the process of crossing the scale that may affect the results, and these assumptions are highlighted in red. (b) Our framework maintains consistency between training and test scales by using degenerate invariance to guide the network to train at original full resolution.**

often at the expense of incompatible spatial and spectral resolutions. It is in this technical context that pansharpening emerged to overcome such limitations. Pansharpening aims to fuse high-resolution panchromatic (HRPAN) and low-resolution multispectral (LRMS) images to produce a high-resolution multispectral (HRMS) composite.

Over decades, extensive research has focused on model-based and deep learning(DL)-based approaches [2, 4, 11, 42, 44] to achieve pansharpening task. Traditional model-based methods require hand-crafted priors to regularize latent solutions, but limited representational ability yields subpar performance on complex scenes. Furthermore, they frequently pose optimization challenges during practical implementation. Deep learning methods demonstrate superiority in representation and generalization compared to handcrafted models. However, existing pansharpening DL method still suffers from a lack of "ground truth" supervision to guide network training, representing a long-standing issue. Overall, while significant progress has been made, further advances in deep pansharpening could help unlock remote sensing's potential through techniques enabling enhanced high-resolution multispectral image reconstruction without reliance on supervised labeling [3, 13, 50].

Traditional deep learning pansharpening approaches synthesize lower-resolution PAN and MS counterparts through techniques like bicubic downsampling of the original HRMS images, as shown in Fig1(a). These downsampled images are then used as pseudo "ground truths" to train networks in a supervised manner. However, the relationship between LRMS and HRMS data is often more complex than simple blurring and resizing [37]. Directly downsampling native imagery via interpolation to create targets is thus not fully representative of real degradation processes. Furthermore, this widely used supervised learning paradigm suffers from a drawback that the whole training process only remains at the scale of reduced resolution, which can not regulate original full resolution performance [12]. The red arrow in Fig.1(a) shows two key assumptions that the existing pansharpening training framework relies on: Spatial degradation can be modeled by bicubic downsampling alone, and networks optimized at reduced scales can be generalized to the native domain. We argue that these simplifying assumptions around scale changes would make it challenging for networks trained only at low resolutions to generate the high-fidelity HRMS images required at the original full resolution for practical applications.

We choose to establish spectral and spatial observation models to simulate the degradation process of remote sensing images, and use the established models to help evaluate the generated images, so as to get rid of the dependence on groundtruth. For efficient and accurate fitting, we analyze the relationship between degradation processes on the basis of theoretical derivation and mathematical analysis (details in Section 3.2). What we found is that the same degradation processes remain consistent at a fixed scale, and this degradation invariance can be used to optimize training strategies. In addition, compared with the existing methods that rely on fixed linear spectral models(refer to Eqs.4 and 5) and rough spatial transformations, we hope to build more accurate models. Inspired by the work of Lu et al. [21], we believe it is possible to introduce physical information into neural networks to simulate spectral transformations. We then introduce operator learning, which bridges gaps in our understanding of real physical models, while accurately modeling spectral degradation processes using powerful fitting capabilities for nonlinear functions. In addition, since the object of learning is changed from function to operator, it has the potential to explore the essential properties of spectral transformation.

Based on the idea of exploring the relationship of the degradation process, we developed an unsupervised pansharpening framework. This framework explores pansharpening by modeling the degradation processes involved. First, In order to model the spectral degradation process, we construct an operator learning model called FadeNet(FDNet), which characterizes the pixel-level spectral relationship between multispectral and panchromatic images independently of resolution, and establish an improved spectral observation model. In addition, we construct Downsample Network(DSNet) based on convolutional layer instead of traditional bicubic interpolation to learn resolution-dependent spatial degradation. Through joint training of FDNet and DSNet directly at the original full resolution, the low-resolution PAN output generated from PAN by DSNet should match that obtained by passing MS through FDNet. After training these networks, their outputs provide consistency constraints during full-resolution pansharpening network optimization, avoiding issues from cross-scale changes.

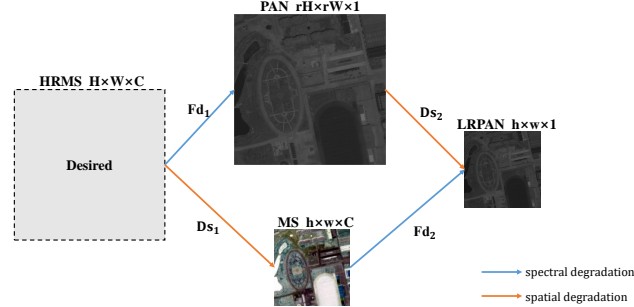

Figure 2: Spatial and spectral degradation framework at the original full resolution. PAN image and LRMS image can be regarded as the degradation results of HRMS image on spectral and spatial scales, respectively, and their continued degradation should result in a consistent low-resolution PAN(LRPAN) image.

Notably, this framework places no constraints on the specific pansharpening model, providing flexibility. As shown in Fig.1, our method can obtain significantly higher quality raw scale HRMS images. To conclude, our contributions include the following three aspects:

- We build a faithful degradation process model and use its invariance to construct consistency constraints. In terms of fitting spectral observation models, to our knowledge, this is the first attempt to introduce operator learning methods to explore the nature of spectral transformations with its ability to accurately model.
- We have built a new unsupervised pansharpening framework that helps existing methods get rid of the limitations of GT by relying solely on MS and PAN images for training, thus obtaining high-quality HRMS images at the original scale.
- Numerous experiments on different satellite datasets show that our method is able to generate higher-quality HRMS images at the original scale without adding any network parameters.

## 2 RELATED WORK

### 2.1 Classic pan-sharpening methods

Traditional pansharpening methods can broadly be categorized into component substitution (CS) based approach [6, 16, 23, 24], multiresolution analysis (MRA) based techniques [22, 27, 34], and other algorithmss [9, 10, 14, 15, 38]. CS techniques are founded on the assumption that the spatial and spectral information of multispectral imagery can be decoupled. With CS, a high-resolution multispectral image is synthesized by combining the spatial detail from a panchromatic image with the spectral content of a low-resolution multispectral counterpart. Over decades, researchers have proposed various decomposition schemes under this paradigm, such as intensity-hue-saturation transformation fusion, Brovey fusion using multiplicative injection, and Gram-Schmidt orthogonalization. However, artifacting may arise if the spectral and spatial information is not separated appropriately. MRA methods apply

multi-scale transforms to panchromatic images to extract spatial details, which are then injected into upsampled low-resolution multispectral imagery. Representative algorithms include high-pass filter fusion and induction fusion. The performance of MRA fusion depends heavily on the choice of multi-scale decomposition. Overall, both CS and MRA fusion have advanced pansharpening but also have certain limitations addressed by more recent DL techniques.

## 2.2 Deep learning based methods

In the last decade, deep learning (DL) methods have been studied for pansharpening, and this type of method directly learns the mapping from LRMS and PAN to HRMS. Typical DL-based pansharpening methods mainly contain two types of network architecture, i.e., residual structure and two-branch structure. The residual structure adds upsampled LRMS images to the output of the network to obtain the HRMS in the form of regression residuals, such as PanNet [44], FusionNet [11], SRPPNN [2], etc [20, 35, 41, 51]. Recently, the two-branch structure is becoming more and more popular. This type of method usually conducts feature extraction for PAN and LRMS image, respectively, and fuses their features to reconstruct HRMS images, such as GPPNN [42], Proximal PanNet [4], etc [1, 5, 40, 43, 53, 54]. Although many unsupervised methods have been proposed [30, 48], most of them still choose to train the network under the downsampling scale. The inaccurate generated images caused by this reason have nothing to do with the design of the network itself, which has become an urgent problem to be solved.

## 2.3 Operator learning methods

In the field of biophysical and biomedical modeling, this synergistic integration between ML tools and multiscale and multiphysics models has been recently advocated. It is widely known that neural networks (NNs) are universal approximators of continuous functions [7, 8, 32]. And a NN with a single hidden layer can accurately approximate any nonlinear continuous operator. The universal approximation theorem of operators is suggestive of the structure and potential of deep neural networks (DNNs) in learning continuous operators or complex systems from streams of scattered data. Thus, the concept of operator learning emerged, which is not only to perform function approximation but to choose to model an entire class of problems in order to obtain a class of general solutions. Compared to other physically combined neural network methods [33], it requires less mathematical theory and more information from the data. Multiple operator networks such as DeepONet [28] and FNO [25] have shown strong performance in several fields [19, 29, 31, 55] and remote sensing is also one of them.

## 3 METHOD

We denoted a pair of PAN and MS images corresponding to the same scene as $\mathbf{P} \in \mathbb{R}^{H \times W \times 1}$ and $\mathbf{M} \in \mathbb{R}^{h \times w \times C}$, and up-sampled MS images to obtain images with the same spatial resolution as $\mathbf{P}$, denoted as $\mathbf{MS} \in \mathbb{R}^{H \times W \times C}$.

### 3.1 Overview

An overview of the proposed approach is given in Fig.3, comprising two main training phases. In the first stage, we leverage degradation

process commutativity to establish a training paradigm at the native scale. Here, the jointly optimized FDNet and DSNet models authentically emulate spectral and spatial transformations, respectively. In the second stage, guided consistency losses are constructed using outputs from the trained degradation models to drive PanNet optimization directly on original data. PanNet can adopt any existing pansharpening architecture.

### 3.2 Degradation Networks

As shown in Fig.2, Let the low-spatial-resolution MS image $\mathbf{M} \in \mathbb{R}^{h \times w \times C}$ and the low-spectral-resolution Pan image $\mathbf{P} \in \mathbb{R}^{H \times W \times 1}$ be the spatially degraded version and spectrally degraded version of ground-truth HRMS image $\mathbf{HM} \in \mathbb{R}^{H \times W \times C}$. Here, $W$, $H$, and $C$ are the width, height and spectral bands of $\mathbf{HM}$ while $w$ and $h$ are the width and height of $\mathbf{M}$, respectively. Then, the degradation models MS and PAN can be modeled as follows:

$$\mathbf{M} = Ds_1 \left( \mathbf{HM} \right), \tag{1}$$

$$\mathbf{P} = Fd_1 \left( \mathbf{HM} \right), \tag{2}$$

where $Ds_1 \left( \cdot \right)$ denote the spatial degradation pipeline, and $Fd_1 \left( \cdot \right)$ is a band-level spectral response in PAN $\mathbf{P}$. With the $\mathbf{M}$ and $\mathbf{P}$, the pansharpening task targets at reconstructing the latent $\mathbf{C}$ of PAN image. However, instead of treating $\mathbf{M}$ and $\mathbf{P}$ as generating objects of the degradation process, we continue to degrade them in the other direction. As shown in Fig.2, we can conclude another set of equations:

$$Fd_2 \left( \mathbf{M} \right) = \mathbf{LP} = Ds_2 \left( \mathbf{P} \right), \tag{3}$$

where $\mathbf{LP} \in \mathbb{R}^{h \times w \times 1}$ means $\mathbf{P}$ is down-sampled with the same spatial resolution of $\mathbf{M}$. $Fd_2 \left( \cdot \right)$ and $Ds_2 \left( \cdot \right)$ can be defined same as $Fd_1 \left( \cdot \right)$ and $Ds_1 \left( \cdot \right)$.

Examining these equations more closely, an intuitive question arises: can the downsampling operations be made mutually consistent? In other words, whether the spectral degradation process and the spatial degradation process have invariance at a fixed scale. To investigate this conjecture, let us analyze the existing formulations of these two degradation processes. Traditionally in pansharpening approaches, a fixed method like simple bicubic resizing is directly applied to model spatial transformations. This approach understandably does not vary with the number of spectral bands, but it may not accurately depict how spatial resolution deteriorates in practice for different sensor types and band configurations. Separately, prevalent assumptions adopted by current spectral observation models aimed at relating multispectral and panchromatic images include:

$$\mathbf{P} = Fd_1 \left( \mathbf{HM} \right) = \sum_{b=1}^{B} \omega_b \cdot \mathbf{M}_b + \epsilon_1, \tag{4}$$

$$\Delta \mathbf{P} = Fd_1 \left( \Delta \mathbf{HM} \right) = \sum_{b=1}^{B} \alpha_b \cdot \Delta \mathbf{M}_b + \epsilon_2, \tag{5}$$

where b is the index of the spectral band, and B is the total number of spectral bands in MS images. $\omega(\cdot)$ and $\alpha(\cdot)$ indicate coefficients of linear combination, and $\epsilon(\cdot)$ is the deviation terms. In other words, these methods consider that the PAN image (or its gradient) can be modeled as a linear combination among all bands (or their gradients) of the MS image. This process is obviously independent

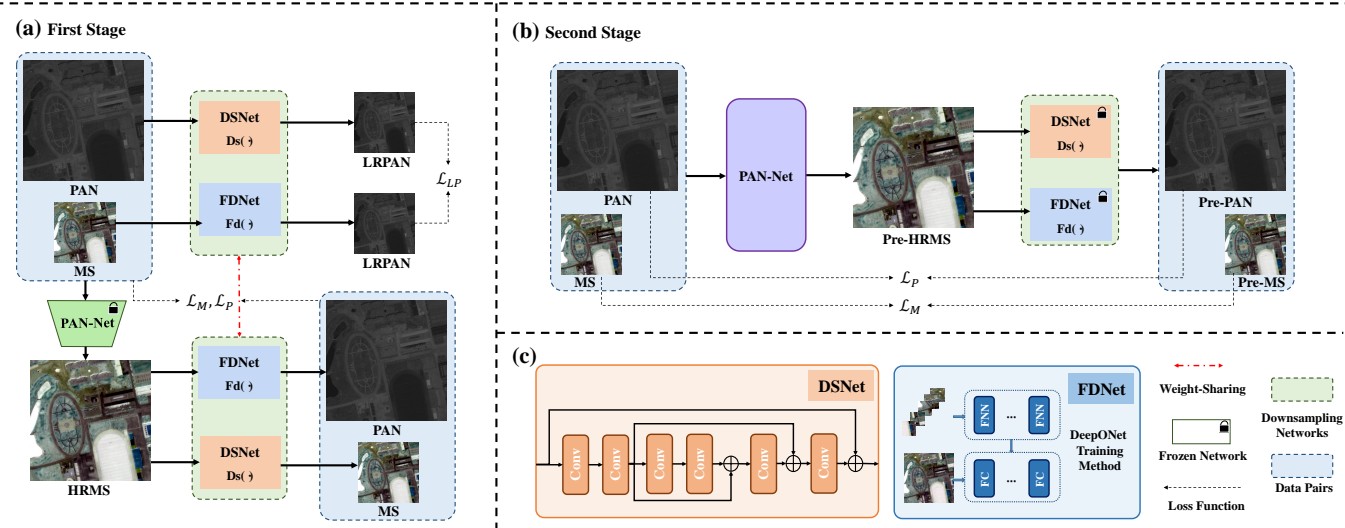

**Figure 3: Overview of our method. (a) The joint training process of FDNet and DSNet. At the original scale, all four degradation processes shown in Fig.2 were simultaneously constructed and network models were designed for learning. Specifically, spatial degradation models are learned using DSNet and spectral observation models are modeled using FDNet. The same type of degradation models share parameters during training. (b) The framework for training the pansharpening network. Using the extracted degradation model, the HRMS image generated by pansharpening network can be degraded into MS image and PAN image, and the loss function can be compared with the input image to realize the training on the original scale. (c) The specific structure of FDNet and DSNet. DSNet is constructed by convolutional layers, and FDNet is based on pixel-level operator learning.**

of the size of the MS image, so the following conclusions can be drawn:

$$Fd_1 (\cdot) = Fd_2 (\cdot), \tag{6}$$

$$Ds_1 (\cdot) = Ds_2 (\cdot), \tag{7}$$

That's what we're trying to prove: the spectral degradation process and the spatial degradation process have invariance at a fixed scale. In fact, the two-level degenerate relationship between the HRMS and LRPAN images in Fig.2 can be written as below:

$$Fd (Ds (\mathbf{HM})) = Ds (Fd (\mathbf{HM})), \tag{8}$$

The above formula indicates that the spectral degradation process and the spatial degradation process conform to the exchange law, and the exchange order does not affect the generated result. To sum up, we design a network to model the spectral observation process. While existing linear models in Eqs.4 and 5 are reasonable starting points, they may not fully capture nonlinear transformations between multispectral and panchromatic data at fine scales. To better model the degradation process, we introduce operator learning methods proposed by DeepONet [28] and design the spectral downsampling network FDNet. DeepONet has shown strong performance fitting nonlinear functions, which we leverage to model the spectral transformation $Fd$ process. To decouple spectral from spatial information and ensure FDNet's scale invariance, we disassemble the image into pixel inputs. In the DeepONet framework, the network is realized using FNN for training. As shown in Fig.3(c), FNN in FDNet takes pixels as inputs during DeepONet optimization.

The convolutional cores enable processing images directly for further training. FDNet was used to test spectral degradation modeling, and the experimental results were shown in Fig.4. The objective of the sub-experiment is to use the spectral corresponding values of each pixel in the MS image to fit the corresponding pixel intensity values of the PAN image. It can be clearly seen that the predicted PAN image pixel values are very similar to the actual values, which shows the effectiveness of modeling using the operator learning method.

For the spatial degradation process, the constructed DSNet consists of several simple convolutional networks, which consist of a convolution operation using Point Spread Function (PSF) and a spatial downsample operation. The structure of DSNet is shown in Fig.3(c).

### 3.3 Joint Training

To train the degradation networks more effectively, we propose a joint learning strategy as shown in Fig.3(a). The MS image $\mathbf{M}$ and PAN image $\mathbf{P}$ are input into FDNet and DSNet respectively for downsampling of spectral information and spatial information. According to Eq.3, both of them should output consistent images $\mathbf{LP}$. Therefore, the consistency relationship between the two output images can be established, and a joint learning framework independent of ground truth can be obtained:

$$\mathcal{L}_{LP} = \|Fd (\mathbf{M}) - Ds (\mathbf{P})\|_1 . \tag{9}$$

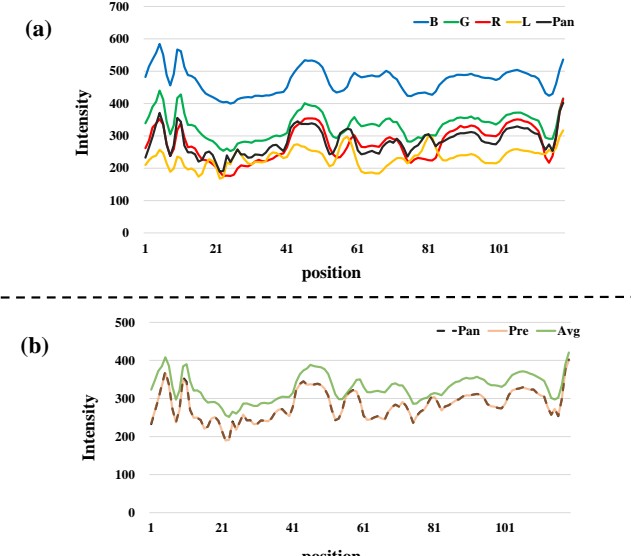

**Figure 4: (a) The intensity values of different spectral bands at corresponding pixel points in MS image and PAN image are correlated. (b) The results are obtained by means of each spectral band and by our operator learning fitting. The predicted results of our method are highly similar to the real values of the corresponding pixels in PAN images.**

However, such constraints are too loose for the training of the network, and the network can easily collapse without getting the desired result. We need to design some other constraints in the loss function to prevent the degradation of the generated image. Given our desire to wean ourselves off ground truth, it's easy to relate to the existing unsupervised pansharpening network.

Introduce a trained unsupervised model $Pr$, then we can get the deduced HRMS image **HM**, using **HM**, we can establish the following two new constraints:

$$\mathcal{L}_P = \|Fd\left(Pr\left(\mathbf{M}, \mathbf{P}\right)\right) - \mathbf{P}\|_1. \tag{10}$$

$$\mathcal{L}_M = \|Ds\left(Pr\left(\mathbf{M}, \mathbf{P}\right)\right) - \mathbf{M}\|_1. \tag{11}$$

Combined with Eqs.9,10 and 11, the loss function of joint training is re-obtained as follows:

$$\mathcal{L}_{joint} = \alpha \mathcal{L}_{LP} + \beta_1 \mathcal{L}_P + \beta_2 \mathcal{L}_M, \tag{12}$$

$\alpha, \beta_1, \beta_2$ are all adjustable hyperparameters.

### 3.4 PanNet Training

Now, with the FDNet and the DSNet, we can train almost any pansharpening network at the original scale as shown in fig.3(b). Specifically, for a PanNet that we want to train, its inputs are **P** and **M**, and its outputs are denoted as **M̂S**, it can be modeled as follows:

$$\mathbf{\hat{M}S} = PanNet(\mathbf{M}, \mathbf{P}). \tag{13}$$

As the generated predictive high-resolution multispectral image, we refer to Fig.2 to derive its corresponding two degraded versions. **M̂S** is applied to the degraded models $Fd\left(\cdot\right)$ and $Ds\left(\cdot\right)$ trained on

this scale, respectively, to generate predicted MS and PAN images **P̂** and **M̂**, namely:

$$\mathbf{\hat{P}} = Fd(\mathbf{\hat{M}S}), \tag{14}$$

$$\mathbf{\hat{M}} = Ds(\mathbf{\hat{M}S}), \tag{15}$$

In the proposed framework, **P̂** and **M̂** represent outputs derived from degrading the high-resolution **M̂S** generated by the trained pansharpening network. Since the spectral and spatial features were separated during previous training of FDNet and DSNet, **P̂** and **M̂** are able to losslessly represent all features of the multispectral imagery. Accordingly, consistency constraints can be formulated using the predicted **P** and **M** results versus the original input data, to guide further training. Specifically, the loss function is constructed as follows:

$$\mathcal{L}_P = \|\mathbf{\hat{P}} - \mathbf{P}\|_1. \tag{16}$$

$$\mathcal{L}_M = \|\mathbf{\hat{M}} - \mathbf{M}\|_1. \tag{17}$$

Combined with Eqs.9,10 and 11, the loss function of joint training is re-obtained as follows:

$$\mathcal{L}_{pan} = \alpha \hat{\mathcal{L}_P} + \beta \hat{\mathcal{L}_M}, \tag{18}$$

where $\alpha$, $\beta$ are adjustable hyperparameters.

## 4 EXPERIMENTS

In this section, we conduct extensive experiments over three satellite image datasets of the WorldView II(WV2) and GaoFen-2(GF2) to evaluate the model performance.

### 4.1 Datasets and benchmark

For each database, PAN images are cropped into patches with the size of $128 \times 128$ pixels while the corresponding MS patches are with the size of $32 \times 32$ pixels. The above generated data was constructed into our training set. To more fully verify the effect of our method, we conduct experiments at both full resolution and downscaling. So we followed the Wald protocol to generate the data set at the lower sampling scale. Specifically, given the MS image $\mathbf{M} \in \mathbb{R}^{H \times W \times C}$ and the PAN image $\mathbf{P} \in \mathbb{R}^{rH \times rW \times 1}$, both of them are downsampled with ratio $\mathbf{r}$, and then are denoted by $\mathbf{m} \in \mathbb{R}^{\frac{H}{r} \times \frac{W}{r} \times C}$ and $\mathbf{p} \in \mathbb{R}^{H \times W \times 1}$ respectively. During the experiments, $\mathbf{m}$ and $\mathbf{p}$ are regarded as the inputs, while $\mathbf{M}$ is the ground truth.

To evaluate the results of our proposed method, several commonly recognized state-of-the-art Pan-sharpening methods are selected, which are classified into two folds: 1) five representative deep-learning based methods, PNN, PANNET [44], MSDCNN [45], SRPPNN, SFIINET [52] and GPPNN [42]; 2) five promising traditional methods, SFIM [27], Brovey [16], GS [24], IHS [17], and GFPCA [26]. Specifically, when we use the proposed framework to train the DL method, we train the degenerate network on the corresponding data set, and then place the corresponding network in the PAN-Net position in Fig.3(b) for training.

### 4.2 Implementation details and metrics

We implement our networks in PyTorch framework on the PC with a single NVIDIA GeForce GTX 3090Ti GPU. In the training phase of the second stage, they are optimized by Adam optimizer over

**Table 1: The non-reference metrics on the full-resolution dataset. The better results for each pair of methods are highlighted in bold.**

| Method | WorldView II | | | GaoFen-2 | | |
|---|---|---|---|---|---|---|
| | $D_\lambda \downarrow$ | $D_s \downarrow$ | QNR↑ | $D_\lambda \downarrow$ | $D_s \downarrow$ | QNR↑ |
| **PNN** | 0.0865 | 0.1343 | 0.8047 | 0.0600 | 0.1276 | 0.8449 |
| **PNN+Ours** | **0.0764** | **0.1104** | **0.8552** | **0.0586** | **0.1082** | **0.8862** |
| **PANNET** | 0.0830 | 0.1499 | 0.8088 | 0.0637 | 0.1224 | 0.8143 |
| **PANNET+Ours** | **0.0779** | **0.1110** | **0.8473** | **0.0531** | **0.1004** | **0.8645** |
| **MSDCNN** | 0.0856 | 0.1375 | 0.8047 | 0.0605 | 0.1057 | 0.8415 |
| **MSDCNN+Ours** | **0.0804** | **0.1181** | **0.8740** | **0.0524** | **0.0934** | **0.8812** |
| **SRPPNN** | 0.0860 | 0.1218 | 0.8060 | 0.0620 | 0.0993 | 0.8049 |
| **SRPPNN+Ours** | **0.0798** | **0.1095** | **0.8613** | **0.0528** | **0.0875** | **0.8613** |
| **GPPNN** | 0.0880 | 0.1523 | 0.7975 | 0.0643 | 0.1633 | 0.7221 |
| **GPPNN+Ours** | **0.0812** | **0.1266** | **0.8240** | **0.0458** | **0.0875** | **0.7603** |
| **FusionNet** | 0.0943 | 0.1576 | 0.8082 | 0.0882 | 0.1644 | 0.7001 |
| **FusionNet+Ours** | **0.0808** | **0.1224** | **0.8185** | **0.0673** | **0.1024** | **0.8278** |

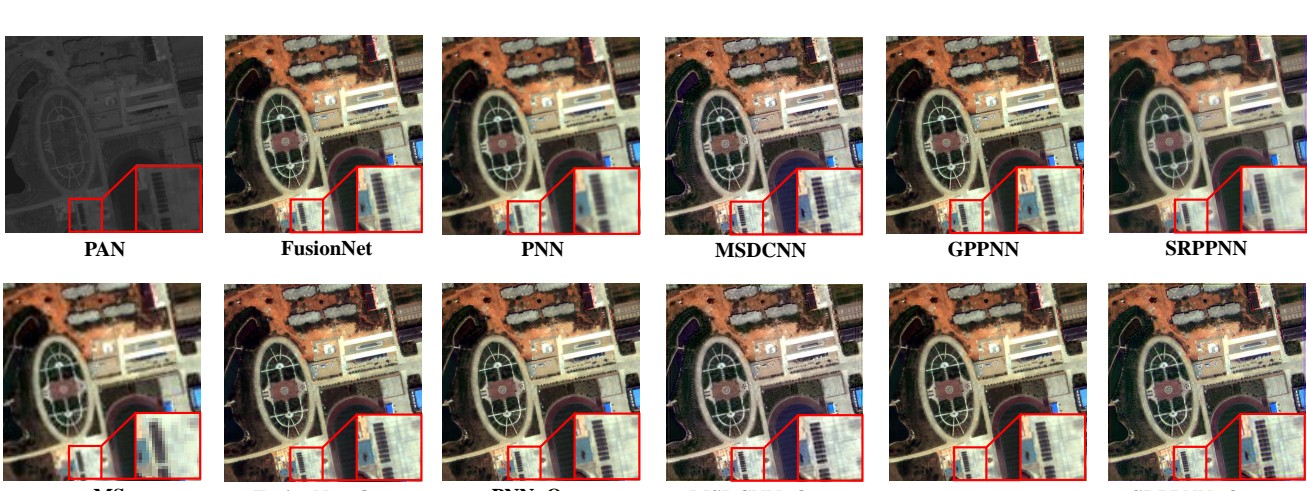

**Figure 5: Visualize comparison of one sample image from the WorldView II dataset.**

500 epochs with a batch size of 16. The learning rate is initialized with $1 \times 10^{-5}$ and decayed by multiplying 0.5 when reaching 200 epochs.

Four popular metrics are used to evaluate the algorithms' performances, including peak signal-to-noise ratio (PSNR) [18], structural similarity (SSIM) [39] and erreur relative global adimensionnelle de synthese (ERGAS) [36] and spectral angle mapper (SAM) [46]. The first three metrics measure the spatial distortion, and the last one measures the spectral distortion. An image is better if its PSNR and SSIM are higher, and ERGAS and SAM are lower. For the full-resolution testing, we adopt the community-popular spectral distortion index quality with no reference (QNR), spectral distortion index ($D_\lambda$), and spatial distortion index (Ds) to evaluate the pan-sharpening performance.

## 4.3 Comparative experiments

In this section, we demonstrate the efficiency of the proposed method on WorldView II and GaoFen-2 and compare it with several methods.

*4.3.1 Qualitative comparison.* We present the qualitative comparison of different methods, which are tested on full-resolution images. Figs.5 and 6 visualize results on different datasets. The first column shows original LRMS and panchromatic inputs. Subsequent columns show pansharpening results from each method, with the first row trained traditionally and the second using our framework. In Fig.5, we choose to zoom in on a set of ordered, tightly packed bars at the bottom of the image. We can see that these tightly packed objects are completely indistinguishable in the MS image, while there are clearer boundaries in the PAN image. In the texture information recovery, the images generated by various methods show obvious differences, the results generated by PNN method can

**Table 2: The performance of a network trained using our method under reduced-resolution testing. The best and second-best results are highlighted in bold and underlined, respectively.**

| Method | WordView II | | | | GaoFen-2 | | | |
|---|---|---|---|---|---|---|---|---|
| | PSNR↑ | SSIM↑ | SAM↓ | EGRAS↓ | PSNR↑ | SSIM↑ | SAM↓ | EGRAS↓ |
| **SFIM** | 34.1297 | 0.8975 | 0.0439 | 2.3449 | 36.906 | 0.8882 | 0.0318 | 1.7398 |
| **Brovey** | 35.8646 | 0.9216 | 0.0403 | 1.8238 | 37.7974 | 0.9026 | 0.0218 | 1.372 |
| **GS** | 35.9376 | 0.9176 | 0.0423 | 1.8774 | 37.226 | 0.9034 | 0.0309 | 1.6736 |
| **IHS** | 35.2926 | 0.9027 | 0.0461 | 2.0278 | 38.1754 | 0.9100 | 0.0243 | 1.5336 |
| **GFPCA** | 34.5581 | 0.9038 | 0.0488 | 2.1411 | 37.9443 | 0.9204 | 0.0314 | 1.5604 |
| **PNN** | 40.7550 | 0.9624 | 0.0259 | 1.0646 | 43.1208 | 0.9704 | 0.0172 | 0.8528 |
| **PANNet** | 40.8176 | 0.9626 | 0.0257 | 1.0557 | 43.0659 | 0.9685 | 0.0178 | 0.8577 |
| **MSDCNN** | 41.3355 | 0.9664 | 0.0242 | 0.9940 | 45.6874 | 0.9827 | 0.0135 | 0.6389 |
| **SRPPNN** | 41.4538 | 0.9679 | 0.0233 | 0.9899 | 47.1998 | **0.9877** | 0.0132 | 0.5586 |
| **GPPNN** | 41.1622 | 0.9684 | 0.0244 | 1.0315 | 44.2145 | 0.9815 | 0.0137 | 0.7361 |
| **SFIINET** | 41.6144 | 0.9689 | 0.0230 | 0.9460 | 47.8541 | **0.9877** | 0.0104 | **0.5191** |
| **MSDCNN+Ours** | 41.6782 | 0.9679 | 0.0230 | 0.9416 | 44.5509 | 0.9750 | 0.0155 | 0.7365 |
| **GPPNN+Ours** | **42.6781** | **0.9710** | **0.0216** | **0.8958** | 47.2752 | 0.9862 | 0.0110 | 0.5594 |
| **SFIINET+Ours** | 41.9304 | 0.9692 | 0.0229 | 0.9324 | **47.8802** | 0.9844 | **0.0102** | 0.5202 |

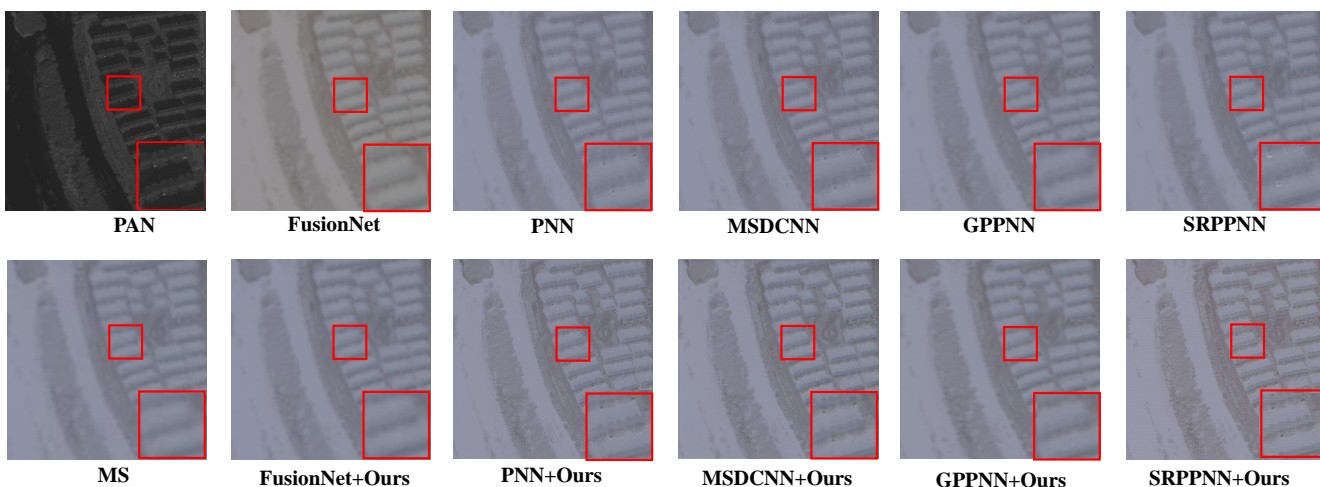

**Figure 6: Visualize comparison of one sample image from the GaoFen-2 dataset.**

hardly recognize the boundary information, and the other methods also have fuzzy phenomena for the restoration of the boundary position. In contrast, by training these methods with our proposed framework, we can see that the generated images are clearly and accurately represented to the object boundaries of our concern. In order to better demonstrate the superiority of our method in texture recovery at full resolution, in Fig.6, we select a scene with snow on the roof of a building for the pansharpening task, in which boundary identification is more difficult. In the test of the original method, most of the methods did not successfully show the boundary of the three segments of color clearly; In tests using our framework training method, this distinction is more easily recognized, especially in the PNN and SRPPNN tests. In summary, our

framework excels at spatial texture and color information recovery for full-resolution pansharpening. Traditional assumptions limit networks' abilities, whereas our approach proves more conducive to generating high-quality HRMS outputs matching native scales. Through improved optimization domain alignment, networks can better reconstruct fine-grained geospatial content critical to remote sensing applications.

*4.3.2 Quantitative comparison.* We further provide quantitative comparisons of these methods on the two datasets from WV2 and GF2. On the one hand, for the no-reference metrics, i.e., $D_\lambda$, $D_s$ and QNR, considering that these metrics do not need reference images and to keep the major advantage of unsupervised methods,

we calculate these metrics on original full resolution images. On the other hand, for the metrics that need the ground-truth data, i.e., ERGAS, PSNR, and SSIM, we downsample the source images into images with a lower resolution and use the original HRMS images as the ground-truth data for calculation. The statistical results of the seven metrics are shown in Table.1 and Table.2.

**Table 3: Performance of different networks on datasets generated using different downsampling methods. Methods with * indicate training using our method. The best and second-best results are highlighted in bold and underlined, respectively.**

| DownSample Method | Pansharpening Method | GaoFen2 | | | |
|---|---|---|---|---|---|
| | | SAM↓ | EGRAS↓ | SSIM↑ | PSNR↑ |
| **Bilinear** | MSDCNN | 0.0159 | 0.7589 | 0.9739 | 44.3015 |
| | GPPNN | 0.0153 | 0.8148 | 0.9760 | 44.0947 |
| | MSDCNN* | 0.0129 | 0.6379 | 0.9806 | 45.9926 |
| | GPPNN* | **0.0116** | 0.5850 | 0.9848 | **46.7786** |
| **Nearest** | MSDCNN | 0.0194 | 0.9748 | 0.9666 | 42.2433 |
| | GPPNN | 0.0183 | 0.9204 | 0.9597 | 42.0749 |
| | MSDCNN* | 0.0132 | 0.6485 | **0.9852** | 46.0560 |
| | GPPNN* | 0.0120 | **0.5731** | 0.9783 | 46.7150 |

In terms of results, in the full-resolution test, our training method can stably obtain better results than the original method on several test data, which is consistent with the visualization results we gave before. Indeed, it can help generate higher quality full-resolution HRMS images, and this improvement has a high degree of generality to various existing methods. In the reduced resolution test, according to the Table.3, our method can achieve a similar level range in results as the baseline and even better, which shows that our training scheme will not cause significant damage to the original network results. Both sets of results show that using our proposed framework to train the network can improve the effect of generating images at full resolution without damaging the network performance under downsampling test setting. In order to further verify our conjecture, we test the network trained in this step on the data set generated by changing the downsampling scheme. As shown in Table.3, due to the influence of spatial downsampling prior on traditional training methods, when the data set generation mode changes, it will be greatly affected, while our method will not be too affected by such interference, which is strong proof that existing training method will learn the bicubic prior.

**Table 4: The pansharpening network trained on the World-View II dataset was tested on the GaoFen-2 dataset. Methods with * indicate training using our method.**

| Method | SAM↓ | EGRAS↓ | SSIM↑ | PSNR↑ |
|---|---|---|---|---|
| GPPNN | 0.121 | 4.235 | 0.542 | 24.225 |
| GPPNN* | 0.040 | 2.259 | 0.928 | 38.556 |

The successful demonstration of operator learning for spectral modeling is certainly encouraging. It prompts us to explore further experiments capitalizing on this promising approach. Specifically,

we augmented training and testing by providing the spectral response function as additional input. In our view, the neural operator should thereby model the sensor's optical properties when optimized across datasets. This would facilitate training a pansharpening network generalizable to new collections. The results in Table.4 validate our hypothesis to an extent, with our framework outperforming traditional techniques significantly.

In conclusion, our method can effectively migrate deep networks to the original full resolution space without significant performance loss on reduced resolution. The network trained by our scheme can generate images with significantly richer and clearer spatial information at the original full resolution. No matter qualitative comparison or quantitative comparison, our proposed method can always generate satisfying performance.

## 4.4 Comparison of efficiency

**Table 5: Efficiency analysis of different methods on the GF2 dataset. Methods with * indicate training using our method.**

| Method | PANNET | PANNET* | SRPPNN | SRPPNN* | GPPNN | GPPNN* |
|---|---|---|---|---|---|---|
| Parameters(M) | 0.233 | 0.233 | 0.342 | 0.342 | 0.275 | 0.275 |
| Running Time(s) | 0.083 | 0.081 | 0.139 | 0.142 | 0.551 | 0.550 |

Our method requires jointly training FDNet and DSNet models tailored to each unique satellite sensor's physical properties and spectral response. This sensor-specific training, alongside generating corresponding MS and PAN images from HRMS inputs using the learned degradation models, increases training time relative to traditional techniques. However, importantly, once optimized our framework incurs no additional parameters or computational cost during inference, as demonstrated by the comparative results in Table.5. While traditional methods approximate sensor traits, our approach more authentically represents real acquisition processes - necessitating upfront optimization efforts but benefiting deployment efficiency in operational use. While sensor specificity elongates initial calibration, the ability to resolve target applications' inherent resolutions justifies this investment.

## 5 CONCLUSION

In this paper, we use the invariance of spectral and spatial degradation processes at the same scale to establish an unsupervised pansharpening network training framework, introduce operator learning and other methods to model the degradation process and do joint training of the two methods, and then use them to train the consistency loss of the network based on the original resolution. Extensive experiments demonstrate our approach stably enhances full-resolution HRMS generation while maintaining network efficacy. Additionally, our framework seamlessly integrates with diverse pansharpening architectures without increasing parameters or degrading inference speed. By enabling resolution-aligned optimization through principled degradation modeling, this work takes a meaningful step toward resolving geospatial details inherently encoded across scales within multispectral imagery. The self-supervised paradigm holds promise to further advance full-fidelity image reconstruction from Earth observation satellites critical to applications in precision agriculture, infrastructure assessment and more.

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
