# OpenReview forum: "Training pansharpening networks at full resolution using degenerate invariance"
_acmmm.org/ACMMM/2024/Conference — MM2024 Oral_

### Official Review · Reviewer_UbVJ · 2024-04-28

**Rating:** 2
**Confidence:** 4

**Summary:**

In this paper, the authors utilize the invariance of spectral and spatial degradation processes at the same scale to establish an unsupervised pan-sharpening network training framework. They train the consistency loss of the network based on the original resolution. Extensive experiments demonstrate their approach stably enhances full-resolution HRMS generation while maintaining network efficacy.

**Strengths:**

1. The authors propose a new unsupervised pan-sharpening framework that does not rely on pseudo-ground truth but uses the invariance of the degradation process to build a consistent loss function on the original scale for network training.
2. The framework diagram of the network is expressed clearly.
3. The method description is detailed.
4. The experimental results confirm the effectiveness of the training strategy proposed in this paper.

**Limitations:**

1. Introducing consistency constraints during the degradation process to implement unsupervised Pan-sharpening is not unprecedented, as demonstrated, for instance, in the paper “Zero-shot semi-supervised learning for pan-sharpening.”
2. Is the assumption that FDNet is equally applicable to both PAN and MS image reasonable? For instance, PAN is a single-channel image while MS has more channels, thus, their spatial degradation processes may not be entirely consistent.
3. Is the design of DSNet and FDNet too simple?

**Suitability:**

3

---

### Official Review · Reviewer_dRpp · 2024-05-24

**Rating:** 5
**Confidence:** 4

**Summary:**

This paper introduces the operator learning method to build an exact mapping function from multi-spectral to panchromatic images and decouple spectral features and texture features. Through joint training, operators and convolutional networks can learn the spatial degradation process and spectral degradation process at full resolution. After training these networks, their outputs provide consistency constraints during full-resolution pansharpening network optimization, avoiding issues from cross-scale changes. The experimental results on different kinds of satellite datasets show that the proposed method outperforms SOTA methods both visually and quantitatively.

**Strengths:**

1. This paper proposes a faithful degradation process model and utilizes its invariance to construct consistency constraints. The authors introduce the operator learning method to explore the nature of spectral transformations with its ability to accurately model.
2. This paper builds a new unsupervised pansharpening framework that helps existing methods get rid of the limitations of GT by relying solely on MS and PAN images for training, thus obtaining high-quality HRMS images at the original scale.
3. Extensive experiments on different satellite datasets show that the proposed method is capable of generating higher-quality HRMS images at the original scale without adding any network parameters.

**Limitations:**

1. Table 2 indicates that the proposed method did not achieve the most advantageous performance in some evaluation metrics. The authors can provide a more detailed and thorough analysis regarding the content of Table 2.
2. The authors should provide relevant experiments or discussions to further analyze the unique advantages of DSNet and FDNet for the pansharpening task, and specifically identify which existing problems they can address.
3. During the Joint Training and PanNet Training processes, the authors could provide more details on the hyperparameter settings of the loss function and indicate whether the method is robust to these hyperparameters.

**Suitability:**

3

---

### Official Review · Reviewer_E9zi · 2024-05-24

**Rating:** 4
**Confidence:** 3

**Summary:**

The paper proposes an interesting hypothesis for the pansharpening task: the spatial and spectral degradations of an image can be separated. Based on this, the authors design a new training paradigm that enables the training of pansharpening networks at full resolution, potentially addressing the poor generalization issue associated with the Wald protocol. The authors validate the effectiveness of the new method on various backbone networks.

**Strengths:**

1. Proposes an interesting new training paradigm that may solve the poor generalization issue of the Wald protocol.
2. Validates the effectiveness of the new method on multiple networks.
3. The paper is logically structured, with clear illustrations and charts to demonstrate the method's workflow.

**Limitations:**

1. The authors' logic regarding the interchangeability of spatial and spectral degradations seems contradictory. In Equations (4) and (5), the authors state that the transformation from hyperspectral image to PAN image can be simply described as a linear transformation. However, Figure 4 suggests that the PAN image differs from the linear combination of MS image channels, and the authors also design a nonlinear neural network, FDNet, to model this transformation. The authors should provide more explanations to address this contradiction.
2. In Equations (10) and (11), the authors state that an unsupervised prior model $Pr$ is needed to train FDNet and DSNet. How is $Pr$ obtained? Does the quality of $Pr$ affect the quality of FDNet and DSNet, thereby impacting the quality of the second stage model training? The authors should discuss $Pr$ in more detail.
3. The authors claim that previous methods typically use simple bicubic resampling to construct reduced-resolution training sets. However, in practice, many datasets[1, 2, 3] are constructed using more complex methods that take into account the modulation transfer function (MTF) of specific satellite sensors on top of polynomial resampling. The statements in Sections 1 and 3.2 may be inaccurate. Furthermore, if the authors incorporate known MTFs into the training of FDNet and DSNet, they may achieve better results.
4. There are issues with details. The notation for image sizes is inconsistent, including in Figure 2, line 300, and line 560. The notation in equations (16)-(18) is inconsistent. Line 548 states that experiments were conducted on three datasets, but only two were actually used.
5. Section 4.4 is somewhat redundant. The paper aims to propose a new method for training networks, and it is evident that a well-trained network will not incur additional overhead.

[1] Liang-Jian Deng, Gemine Vivone, Cheng Jin, and Jocelyn
 Chanussot. Detail Injection-Based Deep Convolutional Neu
ral Networks for Pansharpening. IEEE Transactions on Geo
science and Remote Sensing, 59(8):6995–7010, 2021.

[2] Liang-jian Deng, Gemine Vivone, Mercedes E. Paoletti,
 Giuseppe Scarpa, Jiang He, Yongjun Zhang, Jocelyn
 Chanussot, and Antonio Plaza. Machine Learning in Pan
sharpening: A benchmark, from shallow to deep networks.
 IEEE Geoscience and Remote Sensing Magazine, 10(3):
 279–315, 2022.

[3] Lucien Wald, Thierry Ranchin, and Marc Mangolini. Fusion
 of satellite images of different spatial resolutions: Assessing
 the quality of resulting images. Photogrammetric engineer
ing and remote sensing, 63(6):691–699, 1997.

**Suitability:**

3

---

### Official Review · Reviewer_YDdL · 2024-05-24

**Rating:** 3
**Confidence:** 3

**Summary:**

This paper proposes an unsupervised framework that builds a consistent loss function on the original scale for network training. The consistent loss can be applied to the existing pansharpening method to improve the usability on the original data. Visual and quantitative experimental results have validated the performance of the proposed method. However, there are some problems as follows:
1. In Fig. 3, what is the detailed network structure of PAN-Net? Where are the Fd1, Ds1 and Fd2, Ds2 in Fig. 3?
2. What is the difference between Fd and Ds in Eq. 3?
3. In Eqs. 12 and 13, what are the parameter settings? What is the difference between alpha in Eqs. 12 and 13?
4. As the results are shown in Fig. 6, the superiority of the proposed method is not obvious. More visual results should be presented to validate the performance.
5. More recently proposed (2023) methods should be compared in the experiments.

**Strengths:**

This paper is well-motivated and the idea is somehow interesting. The results have validated the preformance.

**Limitations:**

However, the method is not presented clearly. More comparison experiments should be conducted to demonstrate the effectiveness of the proposed method.

**Suitability:**

2

---

### Meta-Review · Area_Chair_y1RV · 2024-07-04

**Recommendation:** Accept (Oral)
**Confidence:** 4

**Metareview:**

The paper introduces a novel unsupervised framework for pan-sharpening in remote sensing imaging, addressing limitations of existing deep learning methods reliant on pseudo-groundtruth data. By leveraging the invariance of the degradation process, the proposed approach establishes a consistent loss function at the original scale, ensuring high-quality results at full resolution. Key contributions include the introduction of operator learning for precise mapping of multi-spectral to panchromatic images and the decoupling of spectral and texture features. Through joint training, spatial and spectral degradation processes are effectively learned, enhancing the network's performance without relying on downscaled data. Experimental validations across various satellite datasets demonstrate superior visual and quantitative results compared to state-of-the-art methods, underscoring its practical utility and advancement in pan-sharpening technology for remote sensing applications.